# Safety and Preliminary Efficacy of Cervical Paraspinal Interfascial Plane Block for Postoperative Pain after Pediatric Chiari Decompression

**DOI:** 10.3390/healthcare12141426

**Published:** 2024-07-17

**Authors:** Jared M. Pisapia, Tara M. Doherty, Liana Grosinger, Audrey Huang, Carrie R. Muh, Apolonia E. Abramowicz, Jeff L. Xu

**Affiliations:** 1Department of Neurosurgery, Westchester Medical Center, Valhalla, NY 10595, USA; 2School of Medicine, New York Medical College, Valhalla, NY 10595, USA; tara.doherty@wmchealth.org (T.M.D.); lianagrosinger@gmail.com (L.G.); ahuang10@student.nymc.edu (A.H.); apolonia.abramowicz@wmchealth.org (A.E.A.); jeff.xu@wmchealth.org (J.L.X.); 3Department of Anesthesiology, Westchester Medical Center, Valhalla, NY 10595, USA

**Keywords:** cervical cervicis plane (CCeP) block, Chiari type I malformation, pediatric regional anesthesia, suboccipital decompression, intraoperative neuromonitoring

## Abstract

Background: Surgery for lesions of the posterior fossa is associated with significant postoperative pain in pediatric patients related to extensive manipulation of the suboccipital musculature and bone. In this study, we assess the preliminary safety, effect on neuromonitoring, and analgesic efficacy of applying a cervical paraspinal interfascial plane block in pediatric patients undergoing posterior fossa surgery. Methods: In this prospective case series, we enrolled five patients aged 2–18 years undergoing surgery for symptomatic Chiari type I malformation. An ultrasound-guided cervical cervicis plane (CCeP) block was performed prior to the incision. A local anesthetic agent (bupivacaine) and a steroid adjuvant (dexamethasone) were injected into the fascial planes between the cervical semispinalis capitis and cervical semispinalis cervicis muscles at the level of the planned suboccipital decompression and C1 laminectomy. Motor-evoked and somatosensory-evoked potentials were monitored before and after the block. Patients were assessed for complications from the local injection in the intraoperative period and for pain in the postoperative period. Results: No adverse events were noted intraoperatively, and there were no changes in neuromonitoring signals. Pain scores were low in the immediate postoperative period, and rescue medications were minimal. No complaints of incisional pain or need for narcotics were noted at the time of the 3-month postsurgical follow-up. Conclusions: In this study, we demonstrate the preliminary safety and analgesic efficacy of a novel application of a CCeP block to pediatric patients undergoing suboccipital surgery. Larger studies are needed to further validate the use of this block in children.

## 1. Introduction

Surgery for lesions of the posterior fossa is associated with significant postoperative pain secondary to extensive manipulation of the suboccipital musculature and bone. Pain may be more severe when muscles and fascia are directly cut, as in paramedian approaches. Postoperative pain control in pediatric neurosurgical patients undergoing craniotomies may be underestimated [1,2]. Anecdotally, pediatric patients typically minimize overall movement and neck movement for one or more days after surgery. Inadequate postoperative pain control may extend the length of stay and increase associated hospital costs [3]. Pediatric patients are typically treated with a multimodal analgesic regimen after suboccipital surgery [4]. In addition to acetaminophen, anti-inflammatory medications such as ketorolac, muscle relaxants such as diazepam, and narcotics such as oxycodone and morphine are prescribed postoperatively. Recent studies have also demonstrated the potential for liposomal bupivacaine infiltration in postoperative pain management [5]. Enhanced recovery after surgery (ERAS) efforts have recommended reducing narcotic usage due to concerns related to dependency, medication side effects, and blunting of the neurological exam [6]. Thus, there exists an opportunity for improved methods of pain control after posterior fossa surgery in pediatric patients.

The paraspinal interfascial plane block has been studied in adult patients, primarily undergoing spine surgery, and has been associated with decreased postoperative pain and a shortened length of stay [7,8]. The procedure involves using ultrasound guidance to percutaneously deposit a local anesthetic in the space between the muscles adjacent to the spine in order to provide sensory and motor blockade to the dorsal rami of the spinal nerves. Several different types of blocks exist and have recently been classified based on the space targeted between specific muscles [9]. At our institution, the cervical cervicis plane (CCeP) block has shown analgesic efficacy in adult patients; however, the technique has not been assessed in pediatric patients [10].

In this study, we set out to investigate the application of the CCeP block to pediatric patients undergoing posterior fossa surgery. Patients undergoing Chiari I decompression were chosen because Chiari type I malformations are associated with relatively consistent pathology and a standard suboccipital surgical approach. Chiari I malformation is a condition associated with a small posterior fossa in which the lower part of the cerebellum, the cerebellar tonsils, herniate through the foramen magnum and lead to subsequent compression of the brainstem and impairment of cerebrospinal fluid (CSF) flow at the level of the cranio-cervical junction. We examine the safety, effect on neuromonitoring signals, and preliminary analgesic efficacy of the CCeP block in patients undergoing Chiari I decompression.

## 2. Materials and Methods

We prospectively enrolled consecutive patients 2 to 18 years of age with a diagnosis of symptomatic Chiari type I malformation undergoing surgery between 2021 and 2022 at a single academic tertiary care children’s hospital. Patients unable to communicate, thus preventing the acquisition of pain scores, were excluded. Additional exclusion criteria included patients with known bleeding disorders or active scalp infections. Patients with a history of craniotomy within 6 months of Chiari decompression, daily analgesic usage for two weeks prior to surgery, or any surgery or injury requiring analgesic medication within the last 3 months were also excluded. Separate written informed consents for Chiari decompression and for the administration of the CCeP block as part of this study were obtained from the patients or their legal representatives. Institutional IRB approval was obtained for this study.

### 2.1. CCeP Block

Patients underwent a standard induction of general anesthesia with a total intravenous anesthesia technique. A Mayfield skull clamp was applied. Prone positioning was achieved with the use of hip and chest bolsters, ensuring all pressure points were padded. The neck was positioned in flexion such that the suboccipital region was parallel to the floor and at least two fingers fit between the chin and sternum. Intraoperative neuromonitoring electrodes were placed, and baseline motor-evoked potentials (MEPs) and somatosensory-evoked potentials (SSEPs) were obtained. Muscles monitored by needle electromyography (EMG) included the brachioradialis, abductor pollicis brevis, tibialis anterior, and abductor hallucis. Median and posterior tibial nerve SSEPs were monitored as well. Brainstem auditory evoked potentials (BAEPs) were obtained in cases of planned intradural decompression. 

The regional anesthesia team prepped and draped the suboccipital and upper cervical regions. The CCeP block was performed under sterile conditions. The injectate included a combination of bupivacaine 0.25% with epinephrine (1:200,000 up to 1 mL/kg) and 2 mg of dexamethasone. The local anesthetic solution was administered via a 2 inch, 21G, EchoBlock needle (Havel’s, Cincinnati, OH, USA) under direct ultrasound guidance using a 4–12 MHz linear probe (Philips, Bothell, WA, USA) into the desired interfascial space (Figure 1A). The target for anesthetic deposition was the interfascial plane between the cervical semispinalis capitis muscle and the semispinalis cervicis muscle (Figure 1B). The CCeP block was performed above the level of the third cervical vertebrae bilaterally (Figure 1C). 

### 2.2. Surgical Procedure

After the block was completed, the suboccipital and upper cervical regions were again prepped and draped in sterile fashion in preparation for the surgical procedure. The neuromonitoring lead position was confirmed. Repeat MEPs, SSEPs, and BAEPs, when applicable, were obtained after the block and then intermittently throughout the case. An incision extending from the region of the inion to the level of C2 was planned. No additional local anesthetic was given, so as not to confound postoperative pain scores and to avoid potentially toxic levels of the anesthetic being administered. The incision was opened, and dissection proceeded through the midline avascular raphe. A C1 laminectomy and a suboccipital craniectomy were performed. The decision to perform intradural versus extradural decompression was made preoperatively. Criteria for intradural decompression included inferior descent of cerebellar tonsils to the level of C2, rapidly progressive neurologic symptoms or deficits, a history of scoliosis, and the need for occipital cervical fusion. For intradural decompression, a Y-shaped dural opening was made, with the lower limb extending down toward the level of C2. A duraplasty was performed. The wound was closed in a layered fashion. No local anesthetic was given at the end of the case. All patients were sent to the pediatric intensive care unit (PICU) postoperatively. 

### 2.3. Outcome Measures

Patients were evaluated intraoperatively for the safety of the CCeP block and any interference with neuromonitoring signals. The anesthesiologist monitored for changes in blood pressure, heart rate, and electrocardiogram waveform after the block was performed, consistent with current standard practices to assess for signs of local anesthetic toxicity. Prior to the surgical incision, the neurosurgeon and anesthesiologist assessed for signs of hematoma formation after the block was placed. Intraoperative neuromonitoring signals were compared before and 20 min after the block. Signals were evaluated for a 10% increase in latency or a 50% decrease in amplitude. Neuromonitoring was continued until the conclusion of the case. The time from surgery end to extubation was recorded.

In the immediate postoperative period, patients were assessed for complications including pseudomeningocele formation, cerebrospinal fluid leakage, hematoma, and infection. Patients received scheduled oral or parenteral acetaminophen for mild pain. They received oral diazepam and oxycodone as needed for moderate pain and intravenous morphine as needed for severe pain. Pain levels were assessed by nursing staff in the initial 24-h period at specific time points, including upon arrival to the pediatric intensive care unit and at 6, 10, 14, 18, and 24 h after surgery. Age-appropriate pain scales were used, including the Wong–Baker FACES pain rating scale, the FLACC pain scale, and the numeric rating scale [11,12,13,14]. Time to the first rescue dose of pain medication, postoperative opioid consumption, ICU length of stay, and total hospital length of stay were also recorded. 

In the postoperative period, the consenting caregiver completed a questionnaire at 2-week and 3-month follow-up appointments. Data collected included the presence of tenderness or pain at the surgical site, numbness in the area outside of the surgical site, use of postoperative narcotics, complaints regarding the postoperative course, acceptability of pain during the postoperative period, and surgical complications. All data were analyzed using descriptive statistics in Stata (College Station, TX, USA).

## 3. Results

Five pediatric patients (three females and two males, with a mean age of 8.2 years) were enrolled between 2021 and 2022. After the performance of the CCeP block, no vital sign changes were noted to suggest local anesthetic systemic toxicity. No hematoma formation was noted at the injection sites prior to the surgical incision. In addition, no inadvertent intravascular injection of the vertebral artery or subdural space was noted. Neuromonitoring signals were unchanged before and immediately after the CCeP block and remained stable throughout and at the conclusion of all cases. In the immediate postoperative period, no complications were noted. Pain scores were consistently low in the period 10–18 h after the block administration, as expected with the duration of the local anesthetic (Table 1). 

Additionally, in the postoperative period, overall opioid consumption was lower in the 0–12 h following block administration compared to the 13–24 h following block administration (Figure 2). Secondary outcome measures are shown in Table 2. The length of the intensive care unit stay ranged from 1–4 days, and the length of the hospital stay ranged from 1.5–5 days.

At postoperative follow-up visits, none of the patients showed signs of infection, hematoma formation, cerebrospinal fluid leak, pseudomeningocele formation, or neurological deficits. One of the five patients had tenderness over the surgical site during his 2-week visit. He had been discharged without a prescription for diazepam or narcotics. At his 3-month follow-up visit, his incisional pain was resolved, and he was not taking narcotics. At 2-week and 3-month follow-up points, none of the other patients had experienced tenderness or had complaints regarding the postoperative course. There were no signs of a persistent sensory deficit in the suboccipital region after the injection. At the most recent follow-up, none of the five patients had pseudomeningocele, a cerebrospinal fluid leak, a wound hematoma, a wound infection, or a neurological deficit.

## 4. Discussion

In this prospective observational study, we explored the safety of the CCeP block when applied to pediatric patients undergoing Chiari surgery. Our study demonstrates a novel application of a cervical paraspinal interfascial nerve block technique to pediatric patients undergoing posterior fossa surgery. In this pilot study, there were no complications noted in the immediate or delayed postoperative period. No intravascular injection of anesthetic was observed. The vertebral artery is located anterior to the injection site. In addition, several structures separate the injection site from the vertebral artery, including the cervical multifidus muscle, the cervical semispinalis cervicis muscle, and the retrolaminar fascial plane. The use of ultrasound guidance also decreases the risk of an inadvertent injection of a local anesthetic into the vertebral artery. With improved pain control, it is conceivable that patients may be more active in the postoperative period rather than resting and limiting activity secondary to pain, thus potentially predisposing them to increased rates of pseudomeningocele and CSF leak in cases of intradural decompression. However, no patients in this study were found to have any wound healing or CSF-related issues at their most recent follow-up. Furthermore, concern for the blunting of neuromonitoring signals from local anesthetic penetration into the CSF was not supported, as intraoperative neuromonitoring signals were unchanged after the CCeP block was performed. It is biologically plausible that anesthetics are unlikely to diffuse through the fascial layer of the cervical semispinalis cervicis muscle and multifidus muscle and then through the dura to reach the cervical spine or nerve roots to interfere with neuromonitoring signals. Furthermore, these findings are in line with other studies that demonstrated no effect on neuromonitoring signals in adult patients who received the CCeP block prior to cervical laminectomies [10]. The block is performed off of the midline, further reducing the risk of subdural injection.

In addition to establishing the safety of the CCeP block for suboccipital surgery in pediatric patients, our study yielded favorable data on preliminary efficacy with regard to postoperative analgesia. As opposed to infiltrating the skin surrounding the incision, the CCeP block, one of several cervical paraspinal interfascial plane blocks, involves specifically anesthetizing the dorsal rami of spinal nerves and their associated branches that provide sensory innervation to the suboccipital region and surgical site. The nerve block has been used in adult clinical practice for cervical spine surgery, such as decompressive laminectomy and fusion [15]. The current CCeP nerve block used in this study was administered at the third cervical level and represents a cephalad extension of the previously described blocks. We observed that starting from the C3 level, the CCeP block extends up to the inion between the obliquus capitis inferior muscle and semispinalis capitis muscle, where the greater occipital and suboccipital nerves can be blocked [16]. By using ultrasound to precisely target the nerves providing sensation of the suboccipital and upper cervical regions, we noted lower subjective pain scores in the first 10–18 h after the block was administered and then increased when the effects of the local anesthetic were likely to have dissipated. Similarly, we noted that opioid consumption was lower in most patients in the first 0–12 h compared to the 13–24 h following block administration, further supporting the preliminary analgesic efficacy of this block. Subject 3 had the highest pain scores and, correspondingly, had the highest MME consumption.

The CCeP block has several clinical applications. Interfascial plane blocks have been incorporated into routine practice in other specialties, such as the transversus abdominal plane, also known as the TAP block for abdominal surgery [17]. Likewise, in our experience, the CCeP block performed by the regional anesthesia team integrated well into the operating room workflow and did not significantly extend overall procedure time. Although the block can be performed before or after surgery, in the current study, the CCeP block was performed preoperatively once the patient was positioned. A proposed benefit of preoperative administration is that the block is performed when the underlying tissue planes are still intact, thereby facilitating the most accurate delivery of the medication to the desired interfascial target. Another advantage of the CCeP block relates to its versatility in that it can be applied in the treatment of other posterior fossa pathologies and surgical approaches. The midline suboccipital approach used for the Chiari I decompression is a standard surgical technique that can be used to access a variety of intradural posterior fossa lesions, such as brain tumors, vascular malformations, or abscesses. During the approach, dissection occurs through the avascular midline raphe, and paraspinal muscles are swept aside and retracted. The CCeP block may be especially useful for postoperative analgesia in a paramedian approach when an incision is made in the suboccipital region off the midline and dissection proceeds directly through the suboccipital musculature. 

Our study has several limitations. Secondary outcome measures such as time to first rescue medication and length of stay are informative, yet they are difficult to interpret without a comparison group. Comparison to historical controls is challenging in that standardized pain assessments are not commonly used in routine clinical practice, and such measures are retrospective. Furthermore, it can be difficult to compare the perception of pain in adolescents versus young children. In addition, confounding factors may be present that are primarily responsible for secondary outcomes independent of the CCeP block. For instance, the length of stay in a patient with a posterior fossa fourth ventricular tumor may be more driven by the intradural pathology, especially in the presence of neurological deficits or hydrocephalus, than the presence or absence of a CCeP block. Thus, in this study, by including only patients with Chiari type I malformation, efforts were made to minimize confounders related to intradural posterior fossa pathology. Although severity may differ, the surgical approach and intradural pathology in cases of Chiari type I malformation are relatively standardized.

## 5. Conclusions

In this pilot study, we sought to demonstrate that the novel application of a cervical paraspinal interfascial plane block to pediatric patients undergoing suboccipital surgery was not associated with complications in our study or compromise of neuromonitoring signals. Alleviation of pain noted after surgery in patients undergoing the CCeP block supports preliminary analgesic efficacy, although further study is required to support generalizability. Data will be used to support a double-blinded, randomized controlled trial comparing pain control and narcotic consumption in preoperative block versus placebo groups undergoing posterior fossa surgery.

## Figures and Tables

**Figure 1 healthcare-12-01426-f001:**
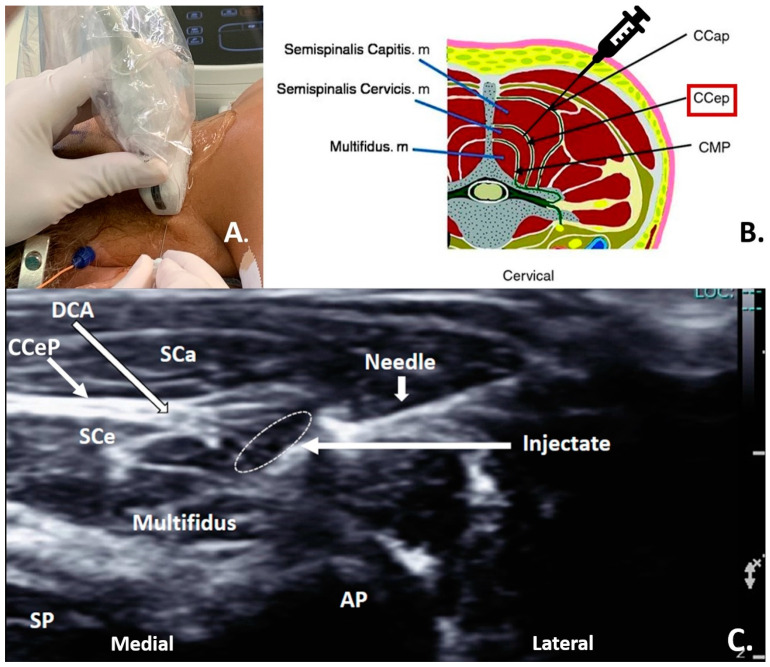
(**A**) Intraoperative ultrasound-guided needle placement for a cervical cervicis plane (CCeP) block. (**B**) Axial diagram showing needle position between semispinalis capitis and cervicis muscles for sensory and motor blockade of cervical nerves. (**C**) Intraoperative transverse ultrasound localizing the third cervical lamina (C3) before injection. CCeP = cervical cervicis plane; CCaP = cervical semispinalis capitis plane; CMP = cervical multifidus plane; SCe = cervical semispinalis cervicis muscle; SCa = cervical semispinalis capitis muscle; Multifiduse = cervical multifidus muscle; DCA = deep cervical artery; SP = spinous process; AP = articular process.

**Figure 2 healthcare-12-01426-f002:**
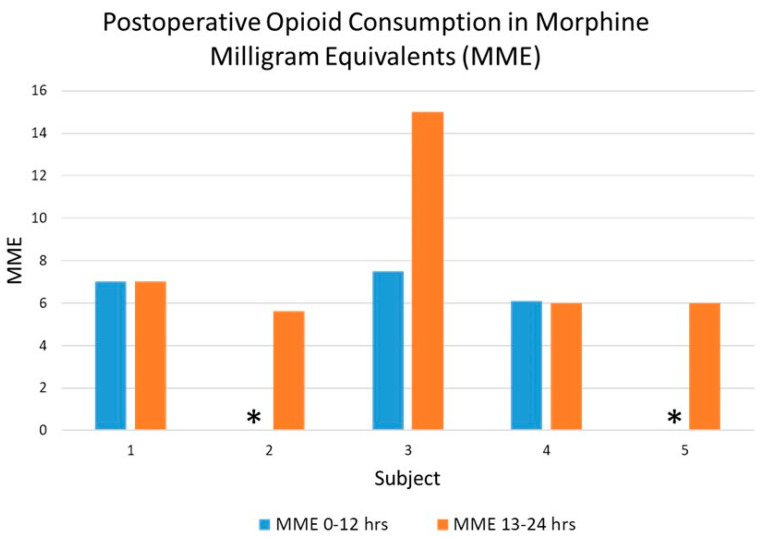
Graph displaying postoperative opioid consumption in the 0–12 h and 13–24 h following block administration. Asterisk (*) = no opioids consumed.

**Table 1 healthcare-12-01426-t001:** Patient age and postoperative pain scores.

ID	Age (Years)	Pain Scoring System	Hour 0	6	10	14	18	24
1	4	FACES	0	9	NR	NR	8	NR
2	4	FLACC	8	0	0	0	4	0
3	16	Numeric	0	4	6	7	6	6
4	5	FACES	4	0	0	0	7	4
5	12	FLACC	0	0	0	0	0	0

The numbers in the column headings refer to hours after surgery. Note: All scales are scored from 1 to 10, with 10 being the worst pain. Abbreviations: NR = not recorded; FLACC = face, legs, activity, cry, and consolability.

**Table 2 healthcare-12-01426-t002:** Surgical approach and secondary outcome measures.

ID	Surgical Approach	Time to Extubation (mins)	Time to Rescue Dose (mins)	ICU LOS (Days)	Hospital LOS (Days)
1	Extradural	8	120	1.5	1.5
2	Intradural	13	83	1	1.5
3	Extradural	20	130	1	3
4	Extradural	NA	40	1	2
5	Intradural	6	508	2	4
Mean +/− SD	-	11.8 +/− 6.2	176.2 +/− 188.8	1.9 +/− 1.2	3.0 +/− 1.5
Median +/− IQR	-	10.5 +/− 9.5	120 +/− 425	1.5 +/− 2	3 +/− 3

Abbreviations: IQR = interquartile range, LOS = length of stay, mins = minutes, SD = standard deviation. Note: One patient remained intubated after surgery for a planned MRI (subject 4, NA).

## Data Availability

The original contributions presented in this study are included in this article. Further inquiries can be directed to the corresponding author.

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
