# Peer review of "Safety and Preliminary Efficacy of Cervical Paraspinal Interfascial Plane Block for Postoperative Pain after Pediatric Chiari Decompression"

_healthcare, 2024, doi:10.3390/healthcare12141426_

Round 1

Reviewer 1 Report

Comments and Suggestions for Authors

A well written article. A few observations are there:

1.      Line 36: Surgery for lesions of the posterior fossa is associated with significant postoperative pain secondary to extensive manipulation of suboccipital musculature and bone. In the abstract this is shortened to posterior cranial fossa surgery. Amendment may be made in the abstract

2.      The third patient had a higher requirement of MMEs and pain score. Time to extubation was more and time to rescue dose was also longer. This has not been explained in the discussion.

3.      Cervical paraspinal interfascial plane block is a deep block with  more severe complications, notably the inadvertent intravascular injection into the vertebral artery. Since the vertebral artery directly supplies blood to the brain, even a minute amount of local anaesthetic can lead to central nervous system (CNS) effects as it swiftly reaches the brain. Another concerning complication of deep CPB is subdural injection. Complications in these 5 patients were not there  but these need to stressed in the discussion

Reviewer 2 Report

Comments and Suggestions for Authors

Dear authors,

Thanks a lot for the submission. This preliminary research well addressed the novel and safe application of a cervical paraspinal interfascial plane block to pediatric patients undergoing suboccipital surgery and this procedure does not compromise neuromonitoring signals in patients with Chiari type I malformation.

However, despite the very limited patient numbers and the lack of a control group, some issues need to be further clarified.

1: In table 1. Patient age and postoperative pain scores, why the Pain scoring system is different for each patient?

2: Should mention or introduce the pain level of postoperative from the normal surgical procedure without Ccep and how long the follow-up time points usually be.

Reviewer 3 Report

Comments and Suggestions for Authors

The work originates from the application to adolescents and children of an infiltration technique on the cervical plane of a local anesthetic in the pre-operative phase, borrowed from a technique applied in adults and which appears to significantly reduce post-operative pain and therefore the consumption of opioids .

The operating technique is well described and the case series, which is very short, is rationally accurate in order to reduce inter-individual variability, which however is maintained with regard to age. This fact, being a study that evaluates pain, is very relevant and associated with the small number of cases makes this study only a preliminary study which signals an interesting intuition, but which requires further confirmation before being able to affirm the validity of the technique.

This limitation is also taken up by the authors who recognize the lack of relevance of a comparison with historical studies, and therefore the need to prepare larger studies, with cohorts that can compare two techniques and that allow age classes to be stratified, since it becomes difficult compare the perception of pain of an adolescent (for example case 3) with that of young children (cases 1,2,4).

Therefore it is necessary to introduce this concept explicitly into the discussion and underline it adequately. Furthermore, a sentence should be added in both the conclusions and the abstract that this is a preliminary study whose conclusions cannot be generalized.

Reviewer 4 Report

Comments and Suggestions for Authors

Thanks to the authors for undertaking this very interesting study on the assessment of safety, effect on neuromonitoring, and preliminary analgesic efficacy when applying a plane block in pediatric patients undergoing posterior fossa surgery.

The paper is very well documented. Also, a thorough explanation of the methodology implemented is given. However, I would like the authors addressing the following:

Specific comments:

1.      Line [166] (Table 1): Could the authors argue as to why (and how) pain scoring systems used are comparable among them?

2.      Line [166] (Table 1): Following point 1 above, could the authors argue why teenagers (12, 16) pain scores are comparable with the ones for children (4,4,5)?

3.      Line [168] (Table 2): Following point 2 above, could the authors argue why the surgical approach is not a variable to factor in when assessing pain?

4.      Line [192]: Prospective observational studies are not able to bring causal conclusions of the form presented here: "...we demonstrated the safety..."

5.      Lines [192] and [210]: Following point 4 above, how the authors objectively measured "safety".

6.      Line [195] I agree with the idea of this being a pilot study (not pilot population). If this was the case, it should have been stated at the beginning of the paper.

7.      Line [260] Could the authors explain the meaning of "case series study" in this context?

8.      Line [264, Conclusion Section]. Is the experiment proposed even feasible? (considering participants’  age range?)

General Comments.

9.       I am very hesitant to approve this study as it is. The main reason is that the quantitative argument used is extremely weak to support the claims given by the authors. That is, I couldn’t find any single piece of evidence about authors attempting to use inferential statistics (although subjects’ sample size would render the calculations useless). 

10.   Although addressed in the abstract, I have also noticed that the discussions and conclusions achieved by the authors do not entirely match the initial premises / research questions.

11. The main issue though, is that I cannot see how it would be possible arguing an effect of the intervention implemented against the absence of control participants. Even though the authors addressed this problem in line 246, the conclusions achieved appear to be causal and need to be changed.

Round 2

Reviewer 4 Report

Comments and Suggestions for Authors

Thanks to the authors for kindly taking the time to review and update the manuscript following reviewers’ comments.

I have not further observations.